# Ftsh Sensitizes Methicillin-Resistant *Staphylococcus aureus* to β-Lactam Antibiotics by Degrading YpfP, a Lipoteichoic Acid Synthesis Enzyme

**DOI:** 10.3390/antibiotics10101198

**Published:** 2021-10-01

**Authors:** Won-Sik Yeo, Bohyun Jeong, Nimat Ullah, Majid Ali Shah, Amjad Ali, Kyeong Kyu Kim, Taeok Bae

**Affiliations:** 1Department of Microbiology and Immunology, Northwest Campus, Indiana University School of Medicine, Indianapolis, IN 46202, USA; wy149@georgetown.edu (W.-S.Y.); majidshah1@yahoo.com (M.A.S.); 2Department of Biology, Georgetown University, Washington, DC 20057, USA; 3Department of Microbiology, College of Medicine, Kosin University, Busan 49267, Korea; ieljbh@gmail.com; 4Atta-ur-Rahman School of Applied Biosciences (ASAB), National University of Sciences & Technology (NUST), Islamabad 44000, Pakistan; nimatscholar@gmail.com (N.U.); amjad.ali@asab.nust.edu.pk (A.A.); 5Department of Precision Medicine, School of Medicine, Sungkyunkwan University, Suwon 16419, Korea; kyeongkyu@skku.edu

**Keywords:** *Staphylococcus aureus*, protease, β-lactam resistance, lipoteichoic acid

## Abstract

In the Gram-positive pathogen *Staphylococcus aureus*, FtsH, a membrane-bound metalloprotease, plays a critical role in bacterial virulence and stress resistance. This protease is also known to sensitize methicillin-resistant *Staphylococcus aureus* (MRSA) to β-lactam antibiotics; however, the molecular mechanism is not known. Here, by the analysis of FtsH substrate mutants, we found that FtsH sensitizes MRSA specifically to β-lactams by degrading YpfP, the enzyme synthesizing the anchor molecule for lipoteichoic acid (LTA). Both the overexpression of FtsH and the disruption of *ypfP*-sensitized MRSA to β-lactams were observed. The knockout mutation in *ftsH* and *ypfP* increased the thickness of the cell wall. The β-lactam sensitization coincided with the production of aberrantly large LTA molecules. The combination of three mutations in the *rpoC*, *vraB,* and SAUSA300_2133 genes blocked the β-lactam-sensitizing effect of FtsH. Murine infection with the *ypfP* mutant could be treated by oxacillin, a β-lactam antibiotic ineffective against MRSA; however, the effective concentration of oxacillin differed depending on the *S. aureus* strain. Our study demonstrated that the β-lactam sensitizing effect of FtsH is due to its digestion of YpfP. It also suggests that the larger LTA molecules are responsible for the β-lactam sensitization phenotype, and YpfP is a viable target for developing novel anti-MRSA drugs.

## 1. Introduction

*Staphylococcus aureus* is a Gram-positive pathogenic bacterium causing multiple diseases ranging from soft-tissue infections to life-threatening diseases such as endocarditis, toxic shock syndrome, and necrotizing pneumonia [1,2]. The methicillin-resistant *S. aureus* (MRSA) infections cannot be effectively treated with most β-lactam antibiotics because it produces PBP2a, a transpeptidase whose active site is not accessible to most β-lactam antibiotics [3,4,5,6,7].

Although PBP2a has been ascribed to the main resistance determinant for β-lactam antibiotics in MRSA, the overall β-lactam resistance of MRSA requires additional molecular factors. For example, since PBP2a can carry out only a transpeptidation reaction, the transglycosylase activity of PBP2 is essential for cell-wall synthesis in MRSA [8]. PBP4 is also required for full resistance to β-lactams [9,10]. The reduced expression of cell-division proteins (e.g., FtsA, FtsW, and FtsZ) sensitizes MRSA to β-lactam antibiotics [11]. Wall teichoic acid (WTA) plays a critical role in the β-lactam resistance of MRSA because it acts as a scaffold to PBP2a by direct binding and is required for the localization of PBP4 [12,13]. Lipoteichoic acid (LTA) is also crucial for MRSA β-lactam resistance [14,15,16,17].

Various mutations are known to increase MRSA β-lactam resistance. By the genome sequencing of a highly resistant subpopulation of four historically early MRSA isolates (i.e., UK13136, ST63/458, E2125, and E4278), the Tomasz group identified 27 mutations [18]. Of the 27 genes, 21 were involved in the induction of stringent stress responses, a known factor to increase MRSA β-lactam resistance [19]; however, the roles of the remaining six genes are unknown. One of them is *ftsH* [18]. FtsH is a membrane-bound ATP-dependent Zn metalloprotease playing critical roles in resistance to various stresses, such as acidic pH, nutritional starvation, and toxic chemicals [20,21]. In *S. aureus*, FtsH directly degrades the following 12 proteins: LrgB, SAUSA300_0310, SAUSA300_0579, SaeQ, YpfP, CydA, CyoE, Ffh, SAUSA300_1351, HemA, HtrA, and MbtS [22,23]. Although the mechanism is unclear, FtsH also represses the transcription of four proteins indirectly: HslO, HrtA, HrtB, and SAUSA300_2637. LrgB is an antiholin-like protein and is involved in autolysis [24,25]. SaeQ is a regulatory protein of the sensor histidine kinase SaeS [26]. YpfP is the enzyme synthesizing the anchor molecule for LTA [27,28]. As cytochrome D ubiquinol oxidase subunit I, CydA is required for respiration [29]. HemA is involved in heme synthesis, whereas CyoE processes heme into the electron transport chain [30]. Ffh is a signal recognition particle protein and exports proteins to the cell membrane [31]. HtrA is a serine protease and plays a role in stress resistance [32]. HslO is predicted to be an Hsp33-like chaperonin, but its function is not defined in *S. aureus*. HrtA and HrtB form a heme-regulated transport system and are involved in heme metabolism [33]. Finally, MbtS is a membrane-bound transcription factor whose physiological function is not completely understood [22].

Independently of the previous study [18], we found that the deletion of *ftsH* increases the β-lactam resistance of *S. aureus* USA300, the predominant MRSA in the US [34]. Since the deletion of *ftsH* increases the MRSA β-lactam resistance, we hypothesized that FtsH overexpression would sensitize MRSA to β-lactams and found that it does. Therefore, in this study, we further sought to identify the underlying molecular mechanism.

## 2. Results

### 2.1. FtsH Sensitizes Methicillin-Resistant S. aureus to Oxacillin

In an agreement with the report by the Tomasz group [18], when we deleted the *ftsH* gene in *S. aureus* USA300, the oxacillin MIC of the strain increased eightfold from 8 µg/mL to 64 µg/mL. The MIC increase was abolished by the further deletion of *mecA* (the gene encoding PBP2a) (Δ*ftsH* vs. Δ*ftsH*Δ*mecA* in Table 1), indicating that PBP2a is essential for the increased oxacillin resistance. Since the deletion of *ftsH* increased the oxacillin MIC, we suspected that the overexpression of FtsH would decrease the MIC. Indeed, when FtsH was overexpressed from pYJ-ftsH-His_6_, a multi-copy plasmid with anhydrotetracycline (ATc)-inducible promoter, the oxacillin MIC of USA300 decreased 16-fold from 8 µg/mL to 0.5 µg/mL (Table 1). Neither the plasmid vector nor the inducer ATc affected the oxacillin MIC (see WT(pYJ335) + ATc in Table 1). Moreover, when FtsH was overexpressed in the hospital-acquired MRSA strain COL and the community-acquired strain MW2 [35,36], their oxacillin MICs also decreased (Table 1). These results indicate that FtsH sensitizes MRSA to β-lactams, and the β-lactam-sensitizing effect of FtsH is likely conserved in *S. aureus* strains.

### 2.2. The Sensitization Effect of ftsH Is Specific to β-Lactams

To explore whether FtsH sensitizes MRSA to other antibiotics, we measured the MIC of various classes of antibiotics for the *ftsH*-deletion mutant of USA300. As shown, the *ftsH*-deletion significantly increased the MIC of β-lactams (i.e., oxacillin, cefotaxime, and cefazolin), which inhibit the transpeptidase activity of PBPs (Table 2). However, it did not significantly alter the MICs of other classes of antibiotics, inhibiting the transglycosylation /transpeptidation step (vancomycin, dalbavancin, and teicoplanin), protein synthesis (linezolid), or cell membrane integrity (daptomycin). Similarly, FtsH overexpression decreased only the MIC of β-lactams (+ ATc in Table 2). These results suggest that the sensitization effect of FtsH is specific to β-lactams.

### 2.3. FtsH Does Not Degrade PBP2a and PBP2

PBP2a and PBP2 are critical for MRSA β-lactam resistance [7,8,37,38]. Similar to FtsH, both PBPs are located in the membrane, raising the possibility that FtsH directly degrades those PBPs, resulting in lower β-lactam resistance. To test this possibility, we measured the expression levels of the PBP2a and PBP2 in WT, Δ*ftsH,* and the FtsH overexpression strain Δ*ftsH* (pYJ-ftsH-His_6_). PBP2a was detected by Western blotting, and PBP2 was visualized by the Bocillin-FL assay, where the fluorescent penicillin is used as a probe to detect PBPs [39]. The FtsH-overexpression strain was grown in the presence or absence of the inducer ATc. As expected, the *ftsH*-deletion increased the cellular levels of the two FtsH substrate proteins, SaeQ and YpfP (Figure 1a). However, similar to the non-substrate protein SrtA, no significant change was observed in the expression level of PBP2a or PBP2 (Figure 1a), indicating that FtsH does not degrade PBP2a and PBP2. When FtsH was overexpressed from pYJ-ftsH-His_6_, the expression level of the FtsH substrate protein SaeQ was decreased; however, no significant changes were observed in the cellular levels of PBP2a or PBP2. Based on these results, we concluded that the β-lactam-sensitizing effect of FtsH is not due to the direct degradation of PBP2a or PBP2 by FtsH.

### 2.4. The Deletion of the FtsH Substrate Gene ypfP Sensitizes the USA300 Strain to Oxacillin

FtsH may sensitize MRSA to β-lactams via those direct or indirect target proteins. To examine this possibility, except for Ffh and SAUSA300_1351, for which no mutant is available, we measured the oxacillin MIC for the rest of the 14 target-gene mutants. As compared with WT (MIC = 8 µg/mL), the *lrgB* and *ypfP* mutants showed decreased oxacillin MIC (*lrgB*, 4 µg/mL; *ypfP*, 0.5 µg/mL) (Appendix A), indicating that the reduction of LrgB and YpfP might be responsible, at least in part, for the β-lactam-sensitizing effect of FtsH overexpression. Due to its greater effect on the oxacillin MIC, the *ypfP* gene was further studied.

### 2.5. The Production of Aberrantly Large LTA Molecules Coincides with Increased Sensitivity to Oxacillin

YpfP is an enzyme catalyzing the last step of the synthesis of diglucosyl diacylglycerol (Glc_2_-DAG), a glycolipid serving as the membrane anchor for LTA [27,28,40]. Since YpfP is the only enzyme synthesizing Glc_2_-DAG, the disruption of *ypfP* abolishes the production of the glycolipid. When that happens, diacylglycerol (DAG), not Glc_2_-DAG, serves as the anchor molecule, resulting in the synthesis of altered LTA [28,41]. Indeed, in the S. aureus USA300 strain, the *ypfP* mutation increased the size of LTA (Δ*ypfP* in Figure 2). In other MRSA strains COL and MW2, the *ypfP* mutation also caused the accumulation of the large LTA (Appendix A). The overexpression of FtsH increased the production of LTA with a slight increase in size, too (Δ*ftsH* (pYJ-ftsH-His_6_) in Figure 2). More importantly, as with the FtsH-overexpression (pYJ-ftsH-His_6_ +ATc in Table 2), the *ypfP* mutation reduced the MIC of β-lactams (Δ*ypfP* in Table 3), which was also reported recently [17]. In the MRSA strains COL and MW2, the *ypfP* mutation also lowered the MIC of three β-lactam antibiotics (i.e., oxacillin, cefotaxime, and cefazolin) by 16 to 2048 times (MIC, 0.125~2 µg/mL depending on antibiotics) (Appendix A). The disruption of the *ypfP* gene in the Δ*ftsH* strain lowered the oxacillin MIC from 64 µg/mL to 0.5 µg/mL, the same as the MIC of the *ypfP* mutant (Appendix A), and increased the production of large LTA (Δ*ftsH::ypfP* in Figure 2). These results suggest that FtsH lowers the β-lactam resistance of MRSA via the degradation of YpfP and that the production of the large LTA might contribute to the reduced β-lactam resistance.

### 2.6. The ftsH and ypfP Mutations Do Not Affect Autolysis Activity but Increase the Cell Wall Thickness

To examine whether the alteration in β-lactam resistance is associated with changes in autolytic activity, we measured the autolytic activity of the WT, *ftsH*, and *ypfP* mutants; however, no significant difference was observed (Appendix A).

Next, since β-lactam antibiotics inhibit bacterial cell wall synthesis, we examined whether the *ftsH* and *ypfP* mutations affect the cell wall thickness of *S. aureus*. The WT, *ftsH*, and *ypfP* mutant strains were grown to exponential growth phase, fixed with glutaraldehyde, and subjected to electron microscopy to measure the cell wall thickness. Because the cell wall thickness was not even, it was measured at four different locations of a cell. About 30 cells of each strain were used for this measurement. As compared with WT, both mutants showed significantly increased cell wall thickness (WT, 26.89 ± 3.95 nm; *ftsH*, 38.6 ± 7.86 nm; *ypfP*, 36.5 ± 6.14 nm, Figure 3). 

### 2.7. Identification of Suppressor Mutations

To further understand the molecular mechanism behind the β-lactam sensitization effects of FtsH, we decided to isolate suppressor mutants of USA300 (pYJ-ftsH-His_6_) whose oxacillin MIC is not decreased by FtsH overexpression. The USA300 (pYJ-ftsH-His_6_) cells were spread on the TSA plate containing 8, 16, and 32 µg/mL oxacillin with ATc (100 ng/mL) and incubated until colonies arose. Colonies grew only on TSA with 8 or 16 µg/mL oxacillin, and the mutant frequency was ~1.4 × 10^−6^ on TSA with 16 µg/mL oxacillin. We sequenced the genomes of 18 suppressor mutants (12 from oxacillin 8 µg/mL and 6 from 16 µg/mL), and the results are shown in Table 4. Intriguingly, every suppressor mutant contained the same mutations in three genes: *rpoC*, *vraB*, and SAUSA300_2133 (hereafter, 2133). The *rpoC* gene encodes an RNA polymerase β’ subunit, whereas the *vraB* gene encodes an acetyl-CoA c-acetyltransferase. The role of the 2133 gene is not known. We named the three mutations ‘core mutations’ (boldfaced in Table 4). Only the core mutations were found in two suppressor mutants, #8-9 and #8-13, indicating that the core mutations are sufficient to overcome the β-lactam sensitizing effect of FtsH. Intriguingly, in the suppressor mutant, the FtsH-overexpression increased LTA production without an apparent size increase (#8-9 in Figure 2 and Appendix A). To assess the effect of the *ypfP* mutation on the oxacillin resistance of the suppressor mutant, we eliminated pYJ-ftsH-His_6_ plasmid from #8-9, resulting in #8-9C, and transduced the transposon insertion mutation of *ypfP* into #8-9C. As shown, the disruption of *ypfP* increased the production of large LTA (#8-9C::*ypfP* in Figure 2) with a concomitant decrease of oxacillin MIC from 32 µg/mL to 0.125 µg/mL (Appendix A). These results suggest that the core mutations are not sufficient to block the β-lactam sensitization effect of the *ypfP*-disruption.

### 2.8. The Suppressor Mutations in vraB and SAUSA300_2133 Confer S. aureus Insensitivity to FtsH

Next, we examined whether all three core mutations are required to confer MRSA insensitivity to FtsH overexpression. Despite repeated trials, we failed to generate the single *rpoC* S723T mutant, suggesting that the mutation alone might be lethal, and mutations in *vraB* and/or 2133 may compensate for such lethality. On the other hand, we successfully created the *vraB* and 2133 single mutations and *vraB*/2133 double mutation in the chromosome of USA300 by allele-replacement [42]. In the oxacillin MIC test, none of the mutations fully restored β-lactam resistance to the level of #8-9 (i.e., 32 µg/mL) (Figure 4a). However, the oxacillin MIC of both mutants was not further reduced by FtsH overexpression (+ in Figure 4a). In addition, unlike WT, the mutants did not show significantly increased LTA production upon FtsH overexpression (Figure 4b). The 2133 mutant produced a higher level of LTA, as compared with WT; however, it was reduced to the WT level by the *vraB* mutation (*vraB*/2133 in Figure 4b).

### 2.9. The Infection by the ypfP Mutant Could Be Treated with Oxacillin

So far, the results are consistent with the notion that FtsH sensitizes MRSA to β-lactam antibiotics by degrading YpfP. This raises the possibility that YpfP can be a target for the development of β-lactam potentiators that convert MRSA to methicillin-sensitive *S. aureus* (MSSA). To test this possibility, we examined whether the murine infection by the *ypfP* mutant can be treated with oxacillin. Although LTA is essential for growth in most *S. aureus,* including USA300, it is not in the MW2 strain [43]. Therefore, in this experiment, we used both the USA300 and the MW2 strains. When the same number of bacterial cells were injected into mice via the retro-orbital route, the *ypfP*-mutant of USA300 showed reduced virulence, compared with WT, whereas such a reduction in virulence was not observed with the *ypfP*-mutant of MW2 (Figure 5a), indicating that the virulence contribution of *ypfP* is strain-dependent. Next, mice were infected with either WT or the *ypfP* mutant via the retro-orbital route and treated with various amounts of oxacillin once per day for three days. Due to the lower virulence of the *ypfP* mutant of USA300, twice more cells were administered to the mice. As expected, regardless of the strain background, the infection by WT strain did not respond to the oxacillin treatment (WT in Figure 5b). However, the *ypfP* mutant infection responded to the oxacillin treatment, although a significantly higher amount was required for the *ypfP* mutant of MW2 (Figure 5b).

## 3. Discussion

As a membrane-bound ATP-dependent protease, FtsH degrades at least 12 membrane and cytoplasmic proteins in *S. aureus* and plays multiple roles in stress resistance, virulence, and β-lactam resistance [18,21,22,23]. In this study, we investigated the molecular mechanism behind the β-lactams sensitization effect of FtsH and found that FtsH sensitizes MRSA to β-lactam antibiotics through the degradation of YpfP, the enzyme synthesizing the membrane anchor molecule for LTA. The higher MIC of the *ftsH* mutant can be explained by the stabilization of YpfP.

The mechanism by which the degradation of YpfP lowers MRSA β-lactam resistance is unclear. The large LTA molecules without the Glc2-DAG anchor are likely the effector molecule bringing about the β-lactam sensitivity to MRSA. For example, under those β-lactam sensitization conditions, MRSA strains produced abnormally large LTA molecules (Figure 2 and Appendix A). In contrast, the suppressor mutant did not show increased production of the large LTA molecules upon FtsH-overexpression (Figure 2). At this moment, it is not clear which of the large size or the absence of the glycolipid anchor molecule contributes more to the β-lactam sensitization effect. Since the effect of FtsH overexpression and the *ypfP*-mutation was specific to β-lactam antibiotics (Table 2), it is possible that the large LTA molecules specifically inhibit the transpeptidase activity of PBPs, in particular, PBP2, PBP2a, and PBP4 that are essential for β-lactam resistance in MRSA [8,9,10]. However, because the LTA synthesis is closely linked to the membrane lipid metabolism [44], the increased synthesis of LTA in the FtsH-overexpression condition is expected to alter the membrane lipid composition, which might indirectly affect the activities of the cell wall synthesis enzymes. These ideas are currently being tested in our laboratory.

The suppressor mutant study showed that the β-lactam-sensitizing effect of FtsH could be blocked by the combination of three core mutations: RpoC S723T, VraB A144S, and SAUSA300_2133 V220L. RpoC is the β’ subunit of the RNA polymerase; therefore, the mutation in RpoC is expected to give a pleiotropic effect on staphylococcal gene transcription and expression. Although the mechanism is not clear, mutations in RNA polymerase subunits are associated with the increase in β-lactam resistance of *S. aureus* [45]. Since we failed to reproduce the mutation in the USA300 genome, we postulate that the mutation alone is lethal to the strain. VraB is an acetyl-CoA c-acetyltransferase, catalyzing the interconversion of acetyl-CoA and acetoacetyl-CoA, and has the potential to affect the synthesis of fatty acid and LTA. In our study, although the *vraB* mutation alone did not greatly affect LTA synthesis, it reduced LTA synthesis in the 2133 mutation (Figure 4b). Unlike RpoC and VraB, the function of 2133 is unknown. According to TMHMM analysis [46], the 2133 protein is a membrane protein with nine transmembrane helices. The Phyre 2 analysis [47] also suggests that the 2133 protein has a significant structural homology (99.4% confidence, 57% coverage) to the concentrative nucleoside transporter from *Vibrio cholerae* [48]. Intriguingly, the *vraB* mutation reduced oxacillin MIC two-fold, whereas the 2133 mutation increased it two-fold, implying that the genes have the opposite effect on β-lactam resistance. However, when the mutations were combined, both MIC and LTA synthesis were similar to those of the *vraB* mutant, demonstrating the dominant role of *vraB* over *2133*. Nonetheless, both the *vraB* and the *2133* mutations limited the increase of LTA synthesis upon FtsH overexpression, which might contribute to the mutants’ insensitivity to the β-lactam sensitization effect of FtsH overexpression. Notably, the MIC of the double mutant (4 µg/mL) is 8 times lower than that of the 8-9 strain (MIC = 32 µg/mL); therefore, it is safe to conclude that the RpoC mutation is the major contributing factor of the increased MIC of the suppressor mutant 8-9.

Previously, Kiriukhin et al. and Ryuno et al. reported that the *ypfP*-knockout mutation increased the LD_50_ of *S. aureus* RN4220 approximately threefold in murine and silkworm infection models, respectively [28,49]. However, in our animal experiment, the *ypfP* mutation decreased the bacterial virulence in USA300 but not in MW2 (Figure 5a), demonstrating that the virulence contribution is strain dependent.

Recently, Hesser et al. reported that the *ypfP* knock-out mutation renders MRSA sensitive to various stressors, including cell wall hydrolases and β-lactam antibiotics [17]. The study also clearly demonstrated that the large LTA molecules, not the absence of YpfP protein, are responsible for the increased sensitivity. In this study, we also showed that, although the extent of sensitivity varied depending on the strains, the infection by *ypfP* mutants can be treated with oxacillin. Therefore, YpfP seems to be a viable target to develop anti-MRSA β-lactam potentiators. If combined with conventional β-lactams such as oxacillin, such drugs are expected to allow the β-lactams to kill MRSA and facilitate the immune-mediated elimination of the bacteria by lowering the bacterial virulence in certain strains.

## 4. Materials and Methods

### 4.1. Bacterial Strains, Plasmids, and Culture Conditions

The bacterial strains and plasmids used in this study are listed in Appendix A [22,23,35,36,42,50,51,52]. Nebraska library strains, whose names start with NE, were acquired from BEI Resources [53]. The ΦΝΞ-8752 is a transposon mutant of *hrtB* in Newman strain [54] and was acquired from Dominique Missiakas’ laboratory at the University of Chicago. *E. coli* and *S. aureus* were grown in lysogeny broth and tryptic soy broth (TSB), respectively. For transduction of mutations and plasmids, heart infusion broth (HIB) with 5 mM CaCl_2_ was used. For MIC measurement, cells were grown in Mueller–Hinton Broth with 2% NaCl. When necessary, antibiotics were added to the growth media at the following concentrations: ampicillin, 100 µg/mL; erythromycin, 10 µg/mL; and chloramphenicol, 5 µg/mL.

### 4.2. Plasmid Constructions and Mutagenesis

For reproducing the mutations in *vraB* and SAUSA300_2133, a 2 kb DNA fragment was PCR-amplified with the primer sets P4203/P4204 (for *vraB*) or with P4205/P4206 (for SAUSA300_2133) (Appendix A). Primers were designed such that the mutation site is located in the middle of the DNA fragment. The DNA fragments were purified and assembled with *Eco*RV-digested pKOR1 by the Gibson method [42,55]. The assembled DNA was inserted into *E. coli*. After confirming the correct insertion of the DNA fragments in pKOR1, the plasmids were electroporated into *S. aureus* RN4220 and subsequently transduced into USA300-P23. The transduced strains were grown in TSB containing chloramphenicol (10 µg/mL, TSB_cm10_) at 30 °C overnight, and the resulting cultures (1–2 µL) were inoculated into 2 mL TSB_cm5_. After overnight growth at 42 °C, the cultures were diluted 10^5^ folds in TSB_cm5_, and the diluted cultures (100 µL each) were spread on TSA_cm5_ and incubated at 42 °C overnight. Five colonies on the plates were grown in TSB_cm5_, and chromosomal DNAs were purified. The co-integration of the plasmid was confirmed by PCR-amplification of the purified DNA with the primer sets P4246/P4248 (for pKOR1-*vraB*) and P4246/P4249 (for pKOR1-2133). The cointegrate strains were grown in TSB at 30 °C overnight. Then the overnight cultures were diluted 100 times in TSB and incubated at 30 °C for 10 h. Finally, the resulting culture was diluted 100 times in TSB and incubate at 30 °C overnight. The cells were diluted 10^5^ times in sterile water, and 100 µL was spread on TSA containing 0.2 µg/mL anhydrotetracycline and incubated at 30 °C overnight. Fifty colonies on the plate were examined for growth in the presence of chloramphenicol (10 µg/mL), and eight colonies, which failed to grow in the presence of chloramphenicol, were selected and grown in TSB at 30 °C. From the chromosomal DNA of the *vraB* and SAUSA300_2133 strains were PCR amplified with the primer sets P4043/P4044 and P4045/P4046, respectively (Appendix A).

### 4.3. Determination of MIC

Minimum inhibitory concentration (MIC) for various antibiotics was determined with a serial dilution of antibiotics according to the recommendations proposed by the Clinical and Laboratory Standards Institute (CLSI) using the microdilution method [56]. Cells were grown in Mueller–Hinton Broth with 2% NaCl. For daptomycin, CaCl_2_ (50 µg/mL final concentration) was added to the media. The MICs of the strains harboring pYJ335 or pYJ-ftsH-His_6_ were assessed in the absence or presence of anhydrotetracycline (100 ng/mL).

### 4.4. Isolation of Suppressor Mutants in the Strain USA Overexpressing FtsH

Overnight culture of USA300Δ*ftsH* carrying pYJ-ftsH-His_6_ was diluted and spread onto TSA containing erythromycin (10 µg/mL), oxacillin (8, 16, and 32 µg/mL), and anhydrotetracycline (100 ng/mL). The plates were incubated at 37 °C until colonies appeared. The colonies were streaked on the same TSA plate to confirm their oxacillin resistance. Then, the FtsH expression was confirmed by Western blotting with an anti-His_6_ antibody. Finally, the suppressor mutants were grown in TSB_erm10_, and genomic DNA was purified with UltraClean Microbial Kit (Qiagen, Hilden, Germany) and sent to the Center for Genomics and Bioinformatics at Indiana University, where the mutant genomes were sequenced with Illumina NextSeq 500.

### 4.5. Western Blot Analysis

Western blot analysis of proteins was carried out as described previously [57]. The SaeS, SrtA, PBP2, and FtsH antibodies were generated by our laboratory. The anti-His_6_ and the anti-PBP2a antibody were purchased from Invitrogen and Abnova, respectively.

Western blot analysis of LTA was carried out as previously described [17], with modifications. Briefly, overnight cultures of the test strains were diluted 1:100 into 10 mL TSB and incubated for 6 h at 37 °C with shaking. When the test strain contains pYJ-ftsH-His_6_, the overnight culture was diluted 1:100 into TSB_erm10_ without or with anhydrotetracycline (100 ng/mL). Cultures were normalized for their optical density (OD_600_ of 4). Cells in the normalized cultures (0.8 mL) were pelleted by centrifugation, suspended in 50 μL of a lysostaphin solution (50 μg/mL lysostaphin, 50 mM Tris, pH 7.4, 150 mM NaCl, 5 mM MgCl_2_), and incubated at 37 °C for 30 min. DNaseⅠ (1 µL, NEB, Ipswich, MA, USA) was added to the cell lysates, and the samples were incubated at 37 °C for 15 min. Then 0.5 μL of proteinase K (20 mg/mL, Invitrogen, Waltham, MA, USA) was added, and the samples were further incubated at 50 °C for 2 h and mixed with 50 μL of 2 × SDS- PAGE sample buffer. Each sample (10 µL) was subjected to 15% SDS-PAGE. Western blotting was performed with anti-LTA monoclonal antibody (Clone 55, HyCult Biotechnology, Wayne, NJ, USA) at a 1:1000 or 1:2000 dilution as primary antibody and the horseradish peroxidase-linked anti-mouse antibody (Cell signaling, Danvers, MA, USA) at a 1:5000 dilution as the secondary antibody.

The Western blots were visualized by the Supersignal West Pico PLUS Chemiluminescent Substrate (Thermo Scientific, Waltham, MA, USA), and the images were taken and processed with LAS-4000 (GE Healthcare, Chicago, IL, USA). All Western blots were repeated at least three times with similar results.

### 4.6. Labeling of Penicillin-Binding Proteins (PBPs) Using Bocillin-FL

Membrane vesicles were prepared as described before [57]. Briefly, overnight cultures of USA300 and USA300Δ*ftsH* were transferred into fresh TSB and incubated for 6 h at 37 °C with shaking. The strain USA300Δ*ftsH* harboring pYJ-ftsH -His_6_ was cultured in TSB_erm10_ without or with anhydrotetracycline (100 ng/mL). Cells were collected by centrifugation, suspended in TSM buffer (20 mM Tris HCl, 0.5 M sucrose, 10 mM MgCl_2_, pH 8.0) containing lysostaphin (50 μg/mL), and incubated at 37 °C for 5 min. After brief sonication of the cell lysates, cell debris was removed by centrifugation at 7000 rpm, 4 °C for 10 min. The membrane fraction was collected by ultracentrifugation (45,000× *g* for 45 min, Beckman Optima Max-XP Ultracentrifuge, Brea, CA, USA). The membranes were washed twice in TKM buffer (50 mM Tris HCl, 50 mM KCl, 1 mM MgCl_2_) and collected by ultracentrifugation (45,000× *g*, 4 °C, 30 min). Finally, the membranes were suspended in TKMG buffer (50 mM Tris HCl, 50 mM KCl, 1 mM MgCl_2_, 25% glycerol, pH 8.0) and stored at −80 °C until use. The membrane vesicles (100 µg) were labeled with 100 µM Bocillin-FL (Molecular Probes, Eugene, OR, USA) for 10 min at 30 °C. The reaction was stopped by adding 5 × SDS-PAGE sample buffer. Labeled membrane proteins were separated on a 7.5% SDS-PAGE gel and detected with a LAS-4000 (GE Healthcare, Chicago, IL, USA) followed by quantification with Multi Gauge software (Fuji Film, V3).

### 4.7. Autolysis Assay

The autolysis assay was carried out as described by Bose et al. [58] without modifications.

### 4.8. Electron Microscopy

The test strains were grown in 3 mL TSB at 37 °C with shaking (200 rpm) to exponential growth phase (OD600 = 1). Cells were collected by centrifugation and suspended in 3% glutaraldehyde (Sigma, St. Louis, MO, USA), 0.1 M sodium phosphate, pH 7.2. After overnight fixation at 4 °C, the samples were sent to the Electron Microscopy Core Facility at Indiana University. At the facility, the specimens were post-fixed in 1% Osmium tetroxide, dehydrated, infiltrated, and embedded in Embed 812 (Electron Microscopy Sciences, Hatfield, PA, USA). Thin sections were cut (80–90 nm), heavy metal stained, then viewed on a Tecnai Spirit (ThermoFisher, Hillsboro, OR, USA). Digital images, including all measurements, were taken with an AMT (Advanced Microscope Techniques, Danvers, MA) CCD camera.

### 4.9. Animal Experiment

The murine blood infection experiment was carried out as described previously with minor modifications [59]. In the experiment, sex-matched C57BL/6J mice (*n* = 10, 8-week-old, Jackson Laboratory) were used. For the virulence test, mice were infected via retro-orbital injection with 10^8^ CFU (USA300 strain) or ~7.5 × 10^7^ CFU (MW2 strain) of either WT or *ypfP* and observed for nine days. For the oxacillin-treatment test for USA300 strain, 10^8^ CFU (USA300 WT) or 2 × 10^8^ CFU (USA300::*ypfP*, MW2 WT, and MW2::*ypfP*) of bacteria were administered into mice via retro-orbital injection. Then, 50 µL of various concentrations of oxacillin (1-28 mg/mL) were administered intramuscularly 1 h after the bacterial infection and daily for two more days. The infected mice were observed for two weeks.

## 5. Conclusions

In *S. aureus*, the ATP-dependent protease FtsH sensitizes the bacterium to β-lactam antibiotics by degrading YpfP, an enzyme synthesizing the glycolipid anchor molecule for LTA synthesis. Since the lack of YpfP increases the size of LTA molecules as well as the thickness of the cell wall, we hypothesize that the large LTA molecules affect the cell-wall synthesis process directly. Finally, the *in vivo* efficacy of oxacillin on the *ypfP* mutants in the murine infection model demonstrates that YpfP is a viable target for the development of β-lactam potentiator against MRSA. 

## Figures and Tables

**Figure 1 antibiotics-10-01198-f001:**
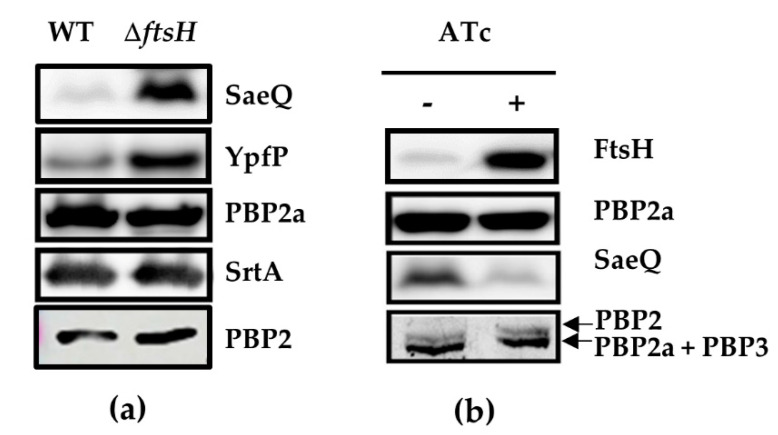
FtsH does not degrade PBP2a and PBP2. (**a**) The effect of the *ftsH*-deletion on the cellular levels of FtsH target proteins and PBPs. Two substrate proteins, SaeQ and YpfP, and the non-substrate protein sortase A (SrtA) were used as positive and negative controls, respectively. YpfP was expressed from pYJ-ypfP-His_6_, whereas all other proteins were from the chromosome. SaeQ, PBP2a, SrtA, and PBP2 were detected with their corresponding antibodies, whereas YpfP was probed with an anti-His_6_ antibody. (**b**) The effect of FtsH-overexpression on the expression levels of PBPs. Δ*ftsH* (pYJ-ftsH-His_6_) was grown with or without ATc. FtsH, PBP2a, and SaeQ were detected by Western blotting, whereas PBP2 was detected by Bocillin-FL assay. ATc, anhydrotetracycline. −, absence; +, presence.

**Figure 2 antibiotics-10-01198-f002:**
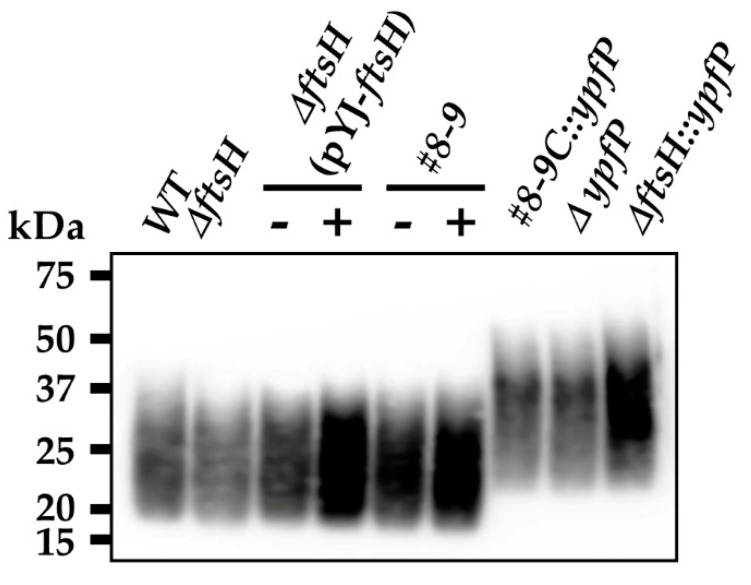
The β-lactam-sensitizing conditions promote the production of large LTA molecules. LTA was extracted from the test strains and separated by 15% SDS-PAGE; then, it was detected by Western blotting with an anti-LTA antibody. WT, wild-type; Δ*ftsH*, *ftsH*-deletion mutant; #8-9, a suppressor mutant of USA300 (pYJ-ftsH-His_6_) that is resistant to the β-lactam sensitizing effect of FtsH; #8-9C, #8-9 without pYJ-ftsH-His_6_; Δ*ypfP*, USA300 with a transposon insertion in *ypfP*; −, no ATc; +, ATc (100 ng/mL).

**Figure 3 antibiotics-10-01198-f003:**
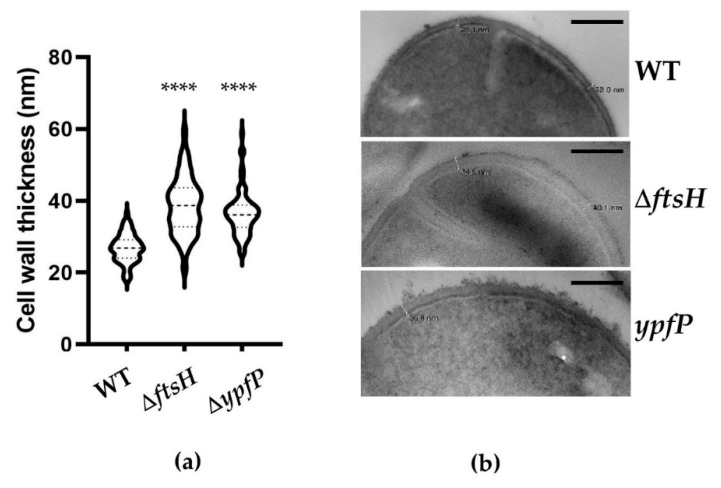
The *ftsH* and the *ypfP* mutations increase the cell wall thickness in USA300. (**a**) The summary of cell wall thickness measurements. The statistical significance of the difference was measured by unpaired Student’s *t*-test. ****, *p* < 0.0001. (**b**) An example of cell wall thickness measurements. The white lines in the bacterial cell images are the measurement points. WT, wild-type; Δ*ftsH*, *ftsH-*deletion mutant; *ypfP*, a transposon-insertion mutant of *ypfP*. Scale bar = 200 nm.

**Figure 4 antibiotics-10-01198-f004:**
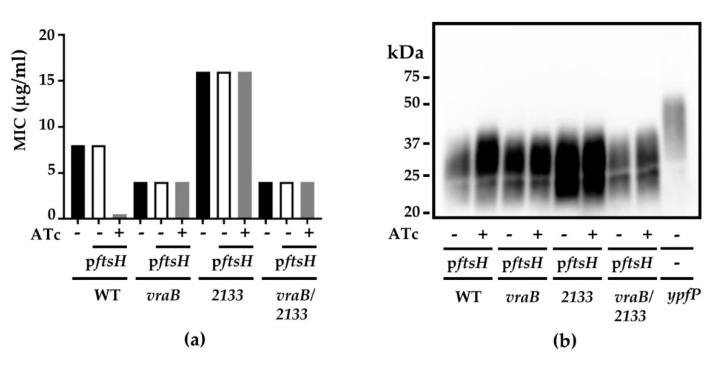
The *vraB* and the *2133* mutations make S. aureus insensitive to the β-lactam sensitizing effect of FtsH overexpression. (**a**) The effect of the *vraB* and the 2133 mutations on oxacillin MIC. MIC was measured in Mueller–Hinton with 2% NaCl with (+) or without (−) FtsH overexpression by anhydrotetracycline (ATc, 100 ng/mL). (**b**) The effect of the *vraB* and the 2133 mutations on the LTA production with (+) or without (−) FtsH overexpression by Atc. Cells were grown in TSB, and LTA was detected by Western blotting with an anti-LTA antibody. p*ftsH*, pYJ-ftsH-His_6_; WT, USA300 wild-type; *vraB*, USA300 with the *vraB* (A144S) mutation; 2133, USA300 with the SAUSA300_2133 (V220L) mutation. *vraB*/2133, USA300 with the double mutation of *vraB* and *2133*.

**Figure 5 antibiotics-10-01198-f005:**
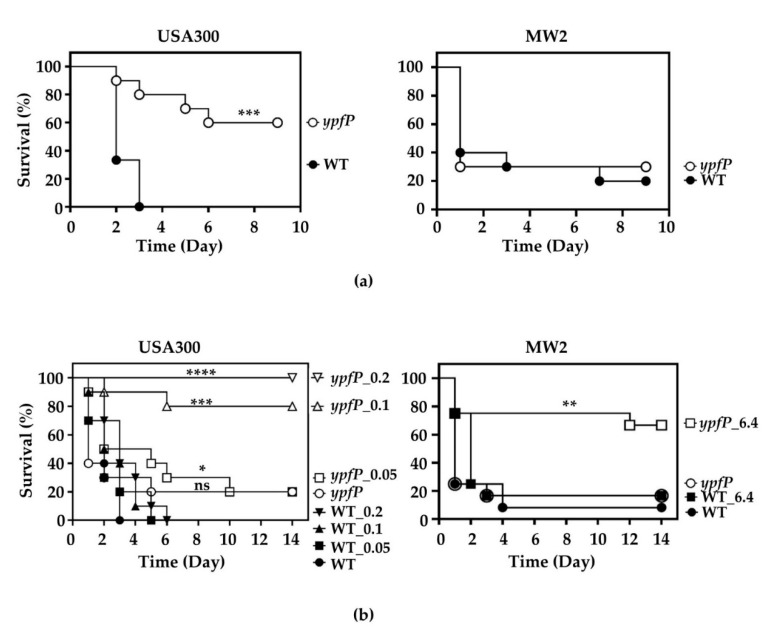
YpfP is a viable target for the development of a β-lactam potentiator. (**a**) The effect of the *ypfP* mutation on staphylococcal virulence. An equal number of the wild-type (WT) or the *ypfP*-mutant cells was injected into 10 mice via the retro-orbital route; then, the infected mice were observed for 9 days. (**b**) The effect of the *ypfP* mutation on oxacillin-treatment. Mice were infected with WT or the *ypfP* mutant via the retro-orbital route; then, the mice were treated with oxacillin via i.m. injection once per day for three days. The mice were observed for two weeks. The numbers on the strain names are the amount of oxacillin used in mg. The statistical significance of murine survival was measured by the Log-rank test. *, *p* < 0.05; **, *p* < 0.005; ***, *p* < 0.001; ****, *p* < 0.0001.

**Table 1 antibiotics-10-01198-t001:** Effect of FtsH on the oxacillin resistance of three MRSA strains.

Strains		MIC (µg/mL)
**USA300**	WT *	8
	Δ*ftsH*	64
	Δ*mecA*	0.25
	Δ*ftsH*Δ*mecA*	0.25
	WT (pYJ335) + ATc **	8
	WT (pYJ-ftsH-His_6_)	8
	WT (pYJ-ftsH-His_6_) + ATc	0.5
**COL**	WT	256
	WT (pYJ335) + ATc	256
	WT (pYJ-ftsH-His_6_)	128
	WT (pYJ-ftsH-His_6_) + ATc	8
**MW2**	WT	8
	WT (pYJ335) + ATc	8
	WT (pYJ-ftsH-His_6_)	4
	WT (pYJ-ftsH-His_6_) + ATc	2

* WT, wild-type. ** ATc, anhydrotetracycline (100 ng/mL).

**Table 2 antibiotics-10-01198-t002:** Effect of FtsH overexpression on the USA300 resistance to various antibiotics.

MIC (µg/mL)				
Antibiotics	WT	Δ*ftsH*	Δ*ftsH*(pYJ-ftsH-His_6_)	
			−ATc	+ATc
Oxacillin	8	64	16	0.5
Cefotaxime	32	128	16	8
Cefazolin	32	128	32	4
Vancomycin	2	1	1	2
Dalbavancin	0.2	0.4	0.4	0.2
Teicoplanin	1	0.5	0.5	1
Linezolid	4	2	2	2
Daptomycin **	0.5	0.25	0.5	0.5

ATc, anhydrotetracycline (100 ng/mL); −, no addition; +, addition. ** Calcium chloride (50 µg/mL) was added to the growth medium.

**Table 3 antibiotics-10-01198-t003:** Effect of the *ypfP* mutation on the USA300 resistance to various antibiotics.

Antibiotics	WT	*ypfP*
Oxacillin	8	0.5
Cefotaxime	32	2
Cefazolin	32	0.5
Vancomycin	2	2
Delbavancin	0.2	0.4
Teicoplanin	1	2
Linezolid	4	1
Daptomycin *	0.5	0.5

* Calcium chloride (50 µg/mL) was added to the growth medium.

**Table 4 antibiotics-10-01198-t004:** Suppressor mutations identified in USA300 (pYJ-ftsH-His_6_).

No.	Gene	Function	Changes in DNA	Changes in Protein	Number of Mutants (Total 18)
1	*rpoC*	RNA polymerase β’ subunit	T591408A	S723T	18
2	*vraB*	Acetyl-CoA c-acetyltransferase	G635085T	A144S	18
3	*2133*	Hypothetical protein	C2306775A	V220L	18
4	*0606*	Hypothetical protein	Insertion A680372 (2 nt) T680375 (4 nt) T680375 (7 nt) T680376 (10 nt)	Frame-shift	3 2 2 2
5	*clpP*	ATP-dependent Clp protease, proteolytic subunit	G838600A C838711A A838825T C839105A A839135G T839161C	E9K A53E I84F A177E D187G Stop196Q	1 1 5 1 2 1
6	*clpX*	ATP-dependent Clp protease, ATP-binding subunit	A1775135C C1775137T	F267V G266D	1 4

## Data Availability

The data presented in this study are available in Appendix A.

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
