# Peer review of "FtsH Sensitizes Methicillin-Resistant Staphylococcus aureus to β-Lactam Antibiotics by Degrading YpfP, a Lipoteichoic Acid Synthesis Enzyme"

_antibiotics, 2021, doi:10.3390/antibiotics10101198_

Round 1

Reviewer 1 Report

The study by Yeo et al. presents a systematic analysis of the importance of FtsH, a membrane-bound ATP-dependent metalloproteinase, to resistance of beta-lactam antibiotics in Staphylococcus aureus. Specifically, the authors document that deletion of the gene ftsH increases the level of beta-lactam resistance in MRSA. An investigation of potential substrates for this protease identified several proteins, of which YpfP was identified as a key instigator in creating an aberrantly large lipoteichoic acid, which increased sensitivity to beta-lactams. YpfP has emerged as a target for inhibition in potentiation of the beta-lactam antibiotics.

The work is solid and will be a nice addition to the literature on MRSA. I recommend its acceptance after a minor revision. Here are a couple of requests for the preparation of a suitable revision:

  1. Lines 36-39. The findings of citation 3 on the allosteric involvement of PBP2a were documented a few years back on structural grounds ( Natl. Acad. Sci. USA, 2013, 110, 16808-16813). Kindly include this citation. The authors should also consider citing as a key recent review this citation (Chem. Rev. 2021, 121, 3412-3463; line 115 and in the Introduction). It discusses many of the introductory issues, including the roles of PBPs and other factors in manifestation of the MRSA phenotype.
  2. The authors have been very good in the correct usage of hyphenation in compound adjectives in many places, but not everywhere. Here are a few (I show the correct usage), where hyphenation was missing in the text: “cell-wall thickness” (line 204), “cell-wall synthesis” (line 208), “cell-wall hydrolases” (line 382) and “cell-wall-synthesis enzymes” (line 349).
  3. Of the two possible mechanistic explanation provided for the effect of FtsH (lines 342-350), I find the first implausible, but the second (the effect on the health of membrane) very plausible. This is especially relevant in the light of documentation that PBP2a (at least) operates in lipid rafts. Nonetheless, the evidence is not in hand one way or another.
  4. I did not like the extensive use of “et al.” in citations. Is this a journal requirement? If not, I prefer full citations with all author names listed.

Author Response

1. Lines 36-39. The findings of citation 3 on the allosteric involvement of PBP2a were documented a few years back on structural grounds ( Natl. Acad. Sci. USA2013110, 16808-16813). Kindly include this citation. The authors should also consider citing as a key recent review this citation (Chem. Rev2021121, 3412-3463; line 115 and in the Introduction). It discusses many of the introductory issues, including the roles of PBPs and other factors in manifestation of the MRSA phenotype.

: Thank you very much for introducing those excellent papers. I was aware of the PNAS paper but not the review paper. I found the review paper very thorough and comprehensive.  Both are very relevant to beta-lactam resistance of MRSA and were added to the places the reviewer suggested (Ref. # 6 and #7)

2. The authors have been very good in the correct usage of hyphenation in compound adjectives in many places, but not everywhere. Here are a few (I show the correct usage), where hyphenation was missing in the text: “cell-wall thickness” (line 204), “cell-wall synthesis” (line 208), “cell-wall hydrolases” (line 382) and “cell-wall-synthesis enzymes” (line 349).

: Those errors were corrected in the revised manuscript (Lines 194, 198, 319, 352).

3. Of the two possible mechanistic explanation provided for the effect of FtsH (lines 342-350), I find the first implausible, but the second (the effect on the health of membrane) very plausible. This is especially relevant in the light of documentation that PBP2a (at least) operates in lipid rafts. Nonetheless, the evidence is not in hand one way or another.

: Thank you for the very thoughtful comment.

4. I did not like the extensive use of “et al.” in citations. Is this a journal requirement? If not, I prefer full citations with all author names listed.

: We did not do it. Endnote did it.

Reviewer 2 Report

First of all, I would like to congratulate you on your work. The study is of interest for the period in which we live. The methods and materials were well chosen. The results of the study are presented clearly, with a sufficient level of detail. The objectives of the study were well defined and significant.

Only minor considerations:

Conclusions: Ideally, at the end to write some more main ideas.

References:  almost 50% of the references are more than 10 years old. Must be updated, if it is possible 

Author Response

Conclusions: Ideally, at the end to write some more main ideas.

: Thank you for your encouragement and the kind comment. The main idea is that the LTA synthesis pathway is linked to beta-lactam resistance of MRSA, and the proteins in the pathway, in particular, the YpfP protein, can be a target for beta-lactam potentiator. We think the last paragraph serves the purpose.

References:  almost 50% of the references are more than 10 years old. Must be updated, if it is possible 

: One of the reasons might be that I usually try to give credit to the original papers. I will be more careful in referencing old papers in the future. 

Reviewer 3 Report

The manuscript by Yeo et al. describes a study to determine the mechanistic link between FtsH and methicillin resistance in S. aureus.  The connection to YpfP is interesting .

Specific Points:

l. 130 Fig. 1 Label in panel B says PBP2a + PBP3. How do you know?  Is PBP2a labelled sufficiently with bocillin?  Why is the same band very dim in panel A?  If YpfP is an FtsH substrate then what happens in the FtsH over-production strain? 

l. 142 YpfP has been shown previously to be required for elevated b-lactam resistance (Hesser et al.). That manuscript needs to be referenced here and say that the phenotype is confirmed.

 l. 190 2.5 and Fig 2 There does not seem to be any correlation between FtsH and LTA size. Why should over-expression of FtsH lead  to an increase in LTA?  If FtsH degrades YpfP then one would expect less LTA.  Previous work has shown that ypfP results in reduced LTA content (Fedtke et al (2007))

Table 4  Is it not surprising to have 3 identical mutations occurring concomitantly in the USA300 (pYJ-ftsH-His6) across 18 isolates?  Does USA300 (pYJ-ftsH-His6) have any SNPs compared to USA300? What genome sequence was used as the template to compare the suppressed isolates?

l. 266 If the rpoC mutation alone is lethal then this should be able to be produced in the double vraB 2133 that was made? Alternatively, the wild type rpoC allele could be knocked into the mutant strain.

Author Response

Thank you so much for your help with our manuscript. For our answer to your comments, please see the file attached.
